# On closed-form tight bounds and approximations for the median of a gamma distribution

**Richard F. Lyon** *

Google Research, Google Inc., Mountain View, California, United States of America

* dicklyon@acm.org

**Data Availability Statement:** All relevant data are within the manuscript.

**Funding:** The author(s) received no specific funding for this work. Author RFL is employed and partially funded by Google. The funder provided support in the form of salary for RFL and the

## Abstract

The median of a gamma distribution, as a function of its shape parameter $k$, has no known representation in terms of elementary functions. In this work we use numerical simulations and asymptotic analyses to bound the median, finding bounds of the form $2^{-1/k}(A + Bk)$, including an upper bound that is tight for low $k$ and a lower bound that is tight for high $k$. These bounds have closed-form expressions for the constant parameters $A$ and $B$, and are valid over the entire range of $k > 0$, staying between 48 and 55 percentile. Furthermore, an interpolation between these bounds yields closed-form expressions that more tightly bound the median, with absolute and relative margins to both upper and lower bounds approaching zero at both low and high values of $k$. These bound results are not supported with analytical proofs, and hence should be regarded as conjectures. Simple approximation expressions between the bounds are also found, including one in closed form that is exact at $k = 1$ and stays between 49.97 and 50.03 percentile.

## Introduction

The gamma distribution PDF is $\frac{1}{\Gamma(k)\theta^k} x^{k-1} e^{-\frac{x}{\theta}}$, but we'll use $\theta = 1$ because both the mean and median simply scale with this parameter. Thus we use this PDF with just the shape parameter $k$, with $k > 0$ and $x \geq 0$:

$$p(k, x) = \frac{1}{\Gamma(k)} x^{k-1} e^{-x}.$$

The mean of this distribution, $\mu$, is well known to be $\mu(k) = k$. The median $v(k)$ is the value of $x$ at which the CDF equals one-half:

$$\frac{1}{2} = \int_0^{v(k)} p(k, x)dx = \int_0^{v(k)} \frac{x^{k-1}}{\Gamma(k)} e^{-x}dx.$$

publication fee for this article, but did not have any additional role in the study design, data collection and analysis, decision to publish, or preparation of the manuscript.

**Competing interests:** Author RFL is employed by Google. This does not alter RFL's adherence to PLOS ONE policies on sharing data and materials. Google has no restrictions on this work.

This equation has no easy solution, but the median is well known to be a bit below the mean, bounded by [1]

$$k - \frac{1}{3} < v(k) < k \quad \text{and} \quad 0 < v(k).$$

Bounds that are tighter in some part of the shape parameter range can be obtained from the known Laurent series partial sums [2, 3], or from the low-$k$ asymptote and bounds of Berg and Pedersen [3].

The Chen and Rubin bounds $k - \frac{1}{3} < v(k) < k$ are tight [1]; the upper bound in the low-$k$ limit and the lower bound in the high-$k$ limit. Recently, Gaunt and Merkle [4] proved that the line of slope 1 that intersects $v(k)$ at the known value $v(1) = \log 2$ is an upper bound for $k \geq 1$; that is, $v(k) < k - 1 + \log 2$, much tighter than the $v(k) < k$ bound, leveraging the prior result at integers by Choi [2] and the result by Berg and Pedersen that the slope of the median $v'(k)$ is everywhere less than 1 [3]. As shown in Fig 1, this new upper bound can be combined with a chord for $0 \leq k \leq 1$, based on convexity shown by Berg and Pedersen [5]. Convexity also implies that any tangent line is a lower bound, and we show later how to find the slope $v'(1)$ at the point where the value $v(1) = \log 2$ is known; that new linear lower bound is included in Fig 1 with the prior linear and piecewise-linear bounds.

A Laurent series for $v(k)$ with rational coefficients has been discovered, with deep connections to some math by Ramanujan. Choi [2] applied Ramanujan's work to this particular question, providing 4 coefficients (through the $k^{-3}$ term). Berg and Pedersen [3], based on work by

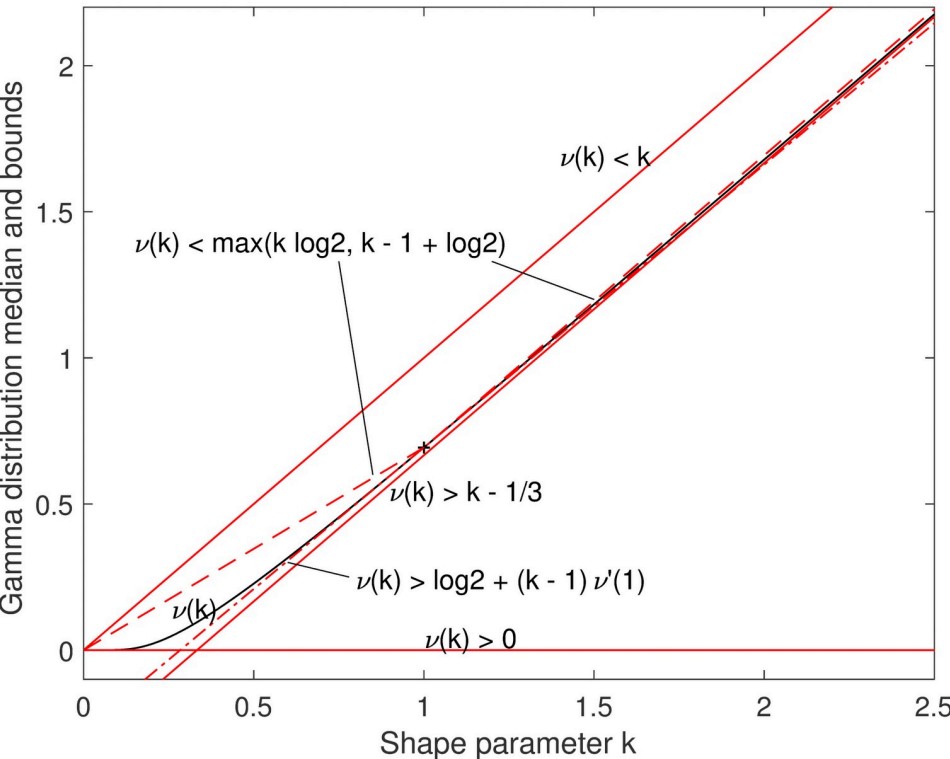

**Fig 1. Linear and piecewise-linear bounds.** The bounds $k - \frac{1}{3} < v(k) < k$ and $v(k) > 0$ (solid lines) are shown along with the true value (solid curve), the piecewise-linear bound that combines the recent linear bound for $k > 1$ [4] with a chord segment (dashed lines), and the linear lower bound that is tangent at $k = 1$ (dash-dot line). The region $k < 1$ is not very usefully bounded.

Marsaglia [6], extended this to 10 coefficients. Neither commented on the radius of convergence, which appears to be in the neighborhood of $k = 1$. For large enough $k$ and $N$, the series yields excellent approximations, but for $k < 1$ it is useless; convergence near $k = 1$ is very slow.

$$v(k) \approx k + \sum_{j=0}^{N} a_j k^{-j}$$

with $a_j = \left\{ \frac{-1}{3}, \frac{2^3}{3^4 \cdot 5}, \frac{2^3 \cdot 23}{3^6 \cdot 5 \cdot 7}, \frac{2^3 \cdot 281}{3^9 \cdot 5^2 \cdot 7}, \frac{-2^3 \cdot 17 \cdot 139753}{3^{13} \cdot 5^3 \cdot 7 \cdot 11}, \frac{-2^3 \cdot 708494947}{3^{15} \cdot 5^3 \cdot 7^2 \cdot 11 \cdot 13} \ldots \right\}$. Thus (where Choi [2] had 144 instead of the correct $2^3 \cdot 23 = 184$):

$$v(k) = k - \frac{1}{3} + \frac{8}{405k} + \frac{184}{25515k^2} + O\left(\frac{1}{k^3}\right).$$

Partial sums of a Laurent series are not generally bounds, but the first two ($k$ and $k - 1/3$) are upper and lower bounds [1], respectively, and the sums ending with $-3$ and $-5$ powers of $k$ also appear (numerically) to be upper and lower bounds, respectively.

Berg and Pedersen [3] also derived an asymptote for small $k$, which we call $v_0(k)$:

$$v(k) \sim v_0(k) = e^{-\gamma} 2^{-1/k},$$

where $\gamma \approx 0.577216$ is the Euler–Mascheroni constant. This asymptote is a lower bound, as we will show. Berg and Pedersen [3] also provide an upper bound $v(k) < e^{-1/3k} k$ that is just above $k - \frac{1}{3}$, and a lower bound $v(k) > 2^{-1/k} k$.

These previous known bounds are illustrated in Fig 2, where upper and lower bounds are distinguished by different line styles. The factor $2^{-1/k}$ is the key to good approximations and bounds, and can be divided out to reduce the dynamic range of values on later plots of $v$ versus $k$. For the range $0.001 < k < 100$, this reduces the dynamic range we need to work with by about 300 orders of magnitude.

Others have shown good bounds and approximations where $k$ is an integer, that is, for the Erlang distribution [2, 7–9]; these do not help us for $0 < k < 1$. The Wilson and Hilferty cube-root transformation [10] of chi-square leads to an approximation at half-integers that is apparently an upper bound: $v(k) < k\left(1 - \frac{1}{9k}\right)^3$ for $k \geq \frac{1}{2}$; if it is, then dropping the final negative term would make it an upper bound for all $k$: $v(k) < k - \frac{1}{3} + \frac{1}{27k}$, which is a bit tighter than the upper bound $v(k) < k - \frac{1}{3} + \frac{1}{18k}$ that Berg and Pedersen [3] proved from their upper bound $v(k) < ke^{-1/3k}$.

We seek upper and lower bounds that are tighter, especially in the middle part of the $k$ range (near $k = 1$), than are previously known. Further, we seek simple approximation formulae for the median, leveraging these bounds via interpolation between them. Main results are summarized in two tables in sections to follow.

The approach of approximating functions by interpolating between upper and lower bounds has been discussed by Barry [11], who used minimax optimization to find numeric parameters in an interpolation-between-bounds approximation to the exponential integral. We are not aware of an interpolation approach being used to find improved bounds in closed form, as we propose here.

## A family of asymptotes and bounds

The Berg and Pedersen lower bounds $2^{-1/k} e^{-\gamma}$ and $2^{-1/k} k$ are tight within their families $2^{-1/k} A$ and $2^{-1/k} Bk$, but not as tight as possible within the wider two-parameter family $2^{-1/k}(A + Bk)$. We find (via numeric and asymptotic evidence, but without proof) that the sum of these two is

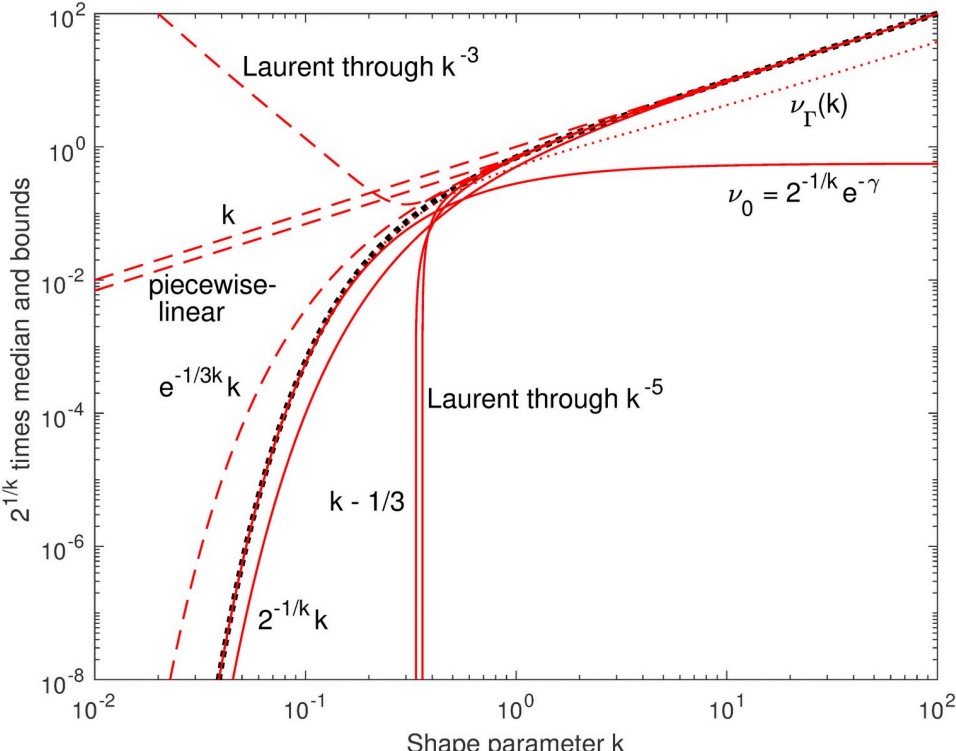

**Fig 2. Previously known bounds.** Previously published bounds (lower bounds solid, upper bounds dashed) for the median of a gamma distribution (heavy dotted), are good at high $k$ or low $k$, but not both. At the left, at $k = 0.01$, the median is near $10^{-30}$. Only $v_0(k)$ is close at low $k$, and it was published as an asymptote, but not a bound; it is strictly less than our new lower bound $v_\Gamma(k)$ (dotted).

the uniquely tight upper bound in that family, and that there is a range of tight lower bounds in the family.

To improve on the lower bounds, and to motivate the family that we consider further, first solve for $v(k)$ in a simple approximation to the distribution's integral, using $e^{-x} < 1$ for $x > 0$:

$$\frac{1}{2} < \int_0^{v(k)} \frac{x^{k-1}}{\Gamma(k)} \, dx$$

$$v_\Gamma(k) = 2^{-1/k} \, \Gamma(k+1)^{1/k} < v(k)$$

which is a tight lower bound, and is a good approximation for $k < 0.1$, but not so great at high $k$—and not what we consider a closed form, due to the gamma function. The factor $2^{-1/k}$, in common to this bound and to Berg and Pedersen's lower bound and asymptote, from a focus on low $k$ instead of the usual high $k$, is key to our approach.

As an aside, since $v_\Gamma(k)$ is a lower bound, we can prove that Berg and Pedersen's asymptote $v_0(k) = e^{-\gamma} 2^{-1/k}$ is a lower bound by showing that $e^{-\gamma} < \Gamma(k+1)^{1/k}$. That $\lim_{k \to 0} \Gamma(k+1)^{1/k} = e^{-\gamma}$ follows from the Taylor series about 0 of $\Gamma(k+1)$, which is $1 - \gamma k + O(k^2)$. That $\Gamma(k+1)^{1/k}$ increases monotonically from there, even though $\Gamma(k+1)$ is decreasing, is proved by showing that its derivative is everywhere positive. Differentiating, in terms of the digamma function

$\psi^{(0)}$, the logarithmic derivative of the gamma function, we find:

$$\frac{d}{dk}\Gamma(k+1)^{1/k} = \frac{\Gamma(k+1)^{1/k}(k\psi^{(0)}(k+1) - \log(\Gamma(k+1)))}{k^2}$$

The only factor here that is not obviously positive for $k > 0$ is $k\psi^{(0)}(k+1) - \log(\Gamma(k+1))$, which at $k = 0$ is equal to 0, and has a surprisingly simple derivative:

$$\frac{d}{dk}\left(k\psi^{(0)}(k+1) - \log(\Gamma(k+1))\right) = k\psi^{(1)}(k+1)$$

Here $\psi^{(1)}$ is the trigamma function, the derivative of the digamma function. This derivative is positive since the trigamma function, a special case of the Hurwitz zeta function, is positive for real arguments, because it has a series expansion with all positive terms:

$$\psi^{(1)}(z) = \sum_{n=0}^{\infty} \frac{1}{(z+n)^2}$$

Therefore $\Gamma(k+1)^{1/k}$ is increasing, as was also obvious numerically, so the Berg and Pedersen asymptote is a lower bound.

The $v_{\Gamma}(k)$ expression resembles the "quantile mechanics" boundary condition for the gamma distribution from Steinbrecher and Shaw [12]. More generally, their work implies that $(u\Gamma(k+1))^{1/k}$ is a lower bound for the $u$ quantile of the gamma distribution, if the coefficients of their power-series recurrence are all positive, for all $0 < u < 1$ and all $k > 0$, as they appear to be. Nevertheless, since they don't claim it as a bound, we say that $v_{\Gamma}(k)$ is new. If their coefficients are all positive, then partial sums of their power series form a family of lower bounds.

The lower bound $v_{\Gamma}(k)$ converges with the Berg and Pedersen asymptote at low $k$. We can improve Berg and Pedersen's asymptote in closed form by utilizing another term. A symbolic calculus system finds for us the next Taylor series terms about $k = 0$ for the power of the gamma function:

$$\Gamma(k+1)^{1/k} \approx e^{-\gamma} + \frac{e^{-\gamma}\pi^2}{12}k - 0.035k^2.$$

With the additional term, we have an improved approximation, which has much less relative error than Berg and Pedersen's at low $k$, and has a high-$k$ behavior nearly proportional to $k$ (but is not a lower bound because we made it larger by ignoring a next negative term):

$$v_1(k) = 2^{-1/k}\left(e^{-\gamma} + \frac{e^{-\gamma}\pi^2}{12}k\right).$$

Inspired by this asymptotic approximation, we consider members of this family of functions, with coefficients $A$ and $B$, and analyze which ones are bounds:

$$\tilde{v}(k) = 2^{-1/k}(A + Bk).$$

Bounds and asymptotic approximations in this family, described in the next section, and a few others are summarized in Table 1. Some of these are the basis for later sections on interpolation between bounds.

## Tight upper and lower bounds

A graphical characterization of this family is most informative. Given some values of $k$ and corresponding numerical $v(k)$, we can find the lines $A + Bk = v(k)2^{1/k}$ in $A$–$B$ space, and plot

**Table 1. Summary of simple bounds and asymptotes.**

| Version | | | Description |
|---|---|---|---|
| $v(k)$ | | | True median of gamma distribution |
| $e^{-1/3k}k$ | | | B&P's upper bound, high-$k$ asymptote |
| $k + \log(2) - 1$ | | | G&M's upper bound for $k \geq 1$ |
| $k \log(2)$ | | | New chord upper bound for $k \leq 1$ |
| $\log 2 + (k-1)v'(1)$ | | | New linear tangent lower bound |
| $v_\Gamma(k) = 2^{-1/k}\,\Gamma(k+1)^{1/k}$ | | | New lower bound, low-$k$ asymptote |
| | $A$ | $B$ | parameters for the form $2^{-1/k}(A + Bk)$ |
| $2^{-1/k}\,k$ | $0$ | $1$ | B&P's lower bound |
| $v_0(k)$ | $e^{-\gamma}$ | $0$ | B&P's low-$k$ asymptote, a lower bound |
| $v_1(k)$ | $e^{-\gamma}$ | $\frac{e^{-\gamma}\pi^2}{12}$ | Improved low-$k$ asymptote; not a bound |
| $v_U(k)$ | $e^{-\gamma}$ | $1$ | New uniquely tight upper bound* |
| $v_{L0}(k)$ | $e^{-\gamma}$ | $0.4596507$ | New tight lower bound*, best at low $k$ |
| $v_{L1}(k)$ | $0.4111107$ | $0.9751836$ | New tight lower bound*, tangent at $k = 1$ |
| $v_{L\infty}(k)$ | $\log\ 2 - \frac{1}{3}$ | $1$ | New tight lower bound*, best at high $k$ |

Comparison of several median bounds and asymptotes. B&P refers to Berg and Pedersen [3], and G&M refers to Gaunt and Merkle [4]. Conjectured bounds that have not been proved, but are supported by asymptotic and numerical results, are marked with an asterisk(*).

them—see Fig 3. Regions full of lines are not bounds, and regions without lines are where bounds are found (including some of Berg and Pedersen's bounds); we're interested in the boundaries between these regions, where tight bounds are to be found.

The improved asymptote $v_1$ is great at low $k$, but is neither an upper nor a lower bound, as shown in Fig 4. We can modify it to approach $k - \frac{1}{3}$ at high $k$ by a few adjustments, via this asymptotic approximation that we get from a symbolic calculus system:

$$k\,2^{-1/k} = k - \log 2 + O(k^{-1}).$$

Thus we find this approximation for high $k$, which appears to be a lower bound as illustrated in Fig 3:

$$v_{L\infty}(k) = 2^{-1/k}\left(\log 2 - \frac{1}{3} + k\right).$$

A compromise approximation for low $k$ mixes these two, differing from the high-$k$ approximation in only the $A$ coefficient, leaving a result consistent to the same order as Berg and Pedersen's asymptote at low $k$, and forming an upper bound as illustrated in Fig 3:

$$v_U(k) = 2^{-1/k}(e^{-\gamma} + k).$$

This mixed approximation has absolute and relative errors approaching zero at low $k$, and relative error approaching zero at high $k$; but the absolute error remains high, near $\log 2 - \frac{1}{3} - e^{-\gamma} \approx 0.20$, at high $k$. These approximations and their errors are illustrated in Fig 5.

These observations are consistent with what Fig 3 suggests: that $A \geq e^{-\gamma} \wedge B \geq 1$ is a necessary and sufficient condition for $2^{-1/k}(A + Bk)$ to be an upper bound to the median, with equality for the tightest upper bound. And for lower bounds, the condition $A \leq e^{-\gamma} \wedge B \leq 1$ is necessary, but not sufficient.

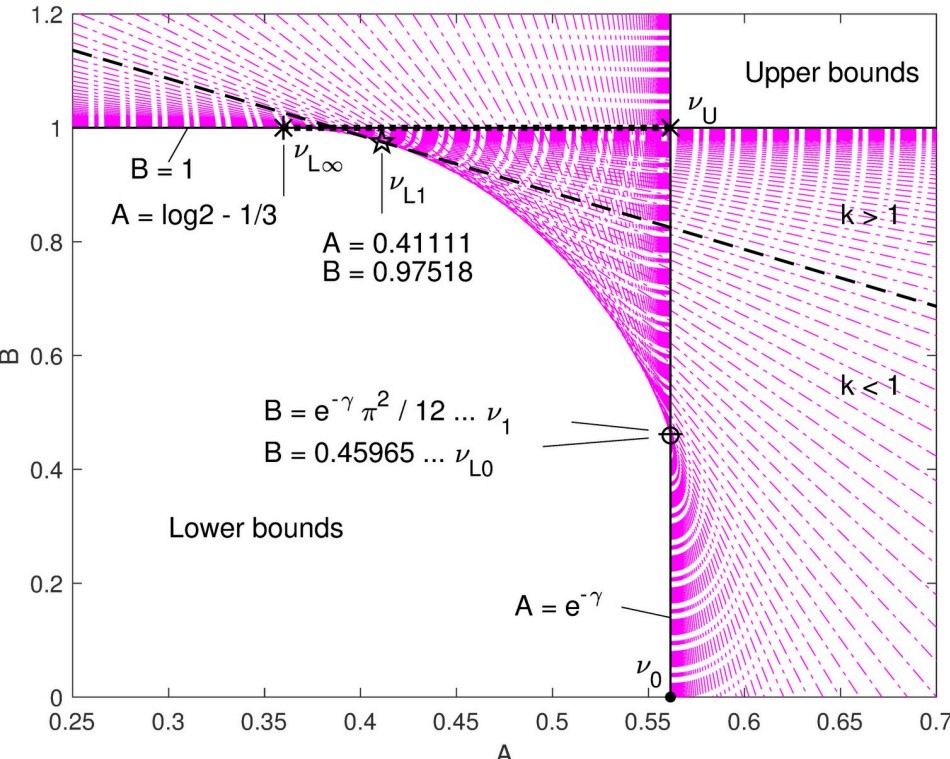

**Fig 3. Parameter space.** The $A$–$B$ parameter space is shaded with dash-dot lines where $v(k)2^{1/k} = A + Bk$, for a set of very small to very large $k$ values in geometric progression (using numerically computed $v(k)$ values). Key values of $A$ and $B$ are indicated. Points outside (or on the edge) of the shaded region represent bounds, while points inside the shaded region represent functions that cross the median function. There is an obvious uniquely tight upper bound $v_U$, and a curved locus of tight lower bounds from $v_{L0}$, which is tightest near $k = 0$, to $v_{L\infty}$, which is tightest for high $k$. One point (pentagram) on the curved locus represents a lower bound $v_{L1}$ that is tight at $k = 1$, for which $A + B = 2\log 2$ (which is the equation of the dashed line). The point $v_1$ represents a good asymptotic approximation close to $v_{L0}$, but not a bound; see the next figure. The dotted line from $v_U$ to $v_{L\infty}$ at $B = 1$ intersects the lines for all $k$ in monotonic order, with $A$ decreasing while $k$ increases.

To support the graphical/numerical observation that $v_{L\infty}(k)$ and $v_U(k)$ are lower and upper bounds, respectively, of the true median ($v_{L\infty}(k) < v(k) < v_U(k)$), we examine their asymptotic behaviors in more detail. At low $k$, it is easy to see, using $\log 2 - \frac{1}{3} \approx 0.359814 < e^{-\gamma} \approx 0.561459$, and $\frac{e^{-\gamma}\pi^2}{12} \approx 0.461781 < 1$, that these differences are positive, for $k \to 0$:

$$2^{1/k}\big(v(k) - v_{L\infty}(k)\big) = e^{-\gamma} - \left(\log 2 - \frac{1}{3}\right) + O(k) > 0.$$

$$2^{1/k}\big(v_U(k) - v(k)\big) = k\left(1 - \frac{e^{-\gamma}\pi^2}{12}\right) + O(k^2) > 0.$$

At high $k$, a symbolic calculus system gives us for $v_{L\infty}(k)$:

$$2^{-1/k}k = k - \log 2 + \frac{\log^2 2}{2} k^{-1} + O(k^{-2})$$

which we can use to construct comparisons to the Laurent series terms for $v(k)$ [2, 3]. Again we find positive differences, with $\frac{\log 2}{3} - \frac{\log^2 2}{2} \approx -0.009177 < \frac{8}{405}$, and $\log 2 - e^{-\gamma} \approx 0.131688 < \frac{1}{3}$,

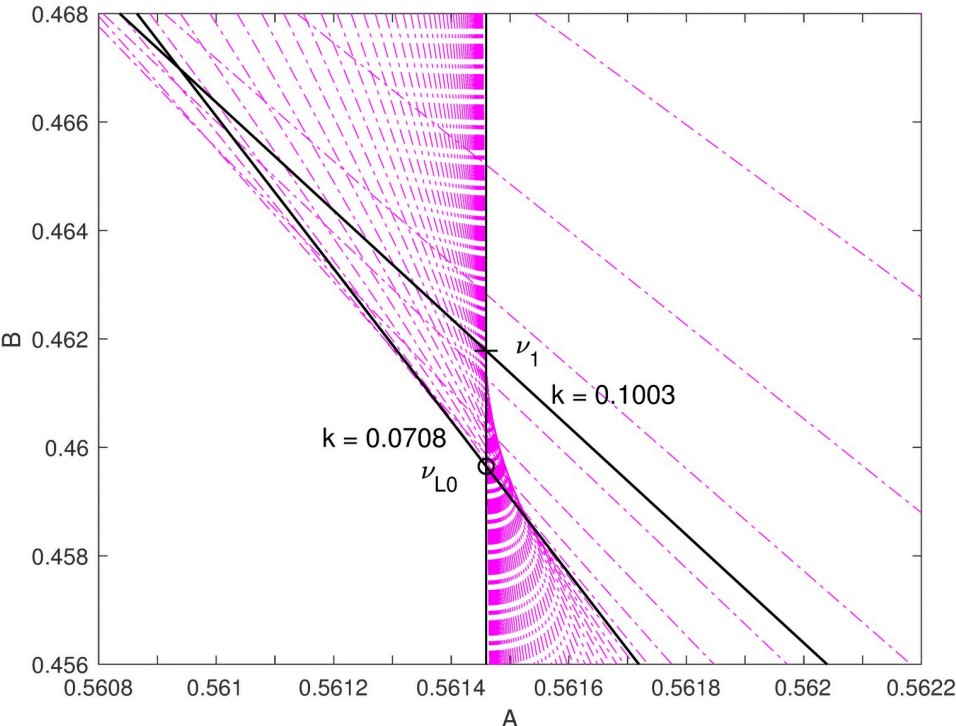

**Fig 4. Parameter space detail.** Zooming in to the parameter neighborhood of $v_1$ and $v_{L0}$, note that the point with $B = e^{-\gamma}\pi^2/12$, which we got from the Taylor series of the power of the gamma function, is actually inside the shaded area, so does not represent a bound; but a point at slightly lower $B = 0.45965$ is on the edge, so represents a lower bound. These points give zero error at approximately $k = 0.1003$ and $k = 0.0708$, respectively (see their errors, or margins, in Fig 5). We do not have analytic formulations for these numeric and graphical observations.

for $k \to +\infty$:

$$v(k) - v_{L\infty}(k) = \left(\frac{8}{405} - \left(\frac{\log 2}{3} - \frac{\log^2 2}{2}\right)\right)k^{-1} + O(k^{-2}) > 0;$$

$$v_U(k) - v(k) = e^{-\gamma} - \log 2 + \frac{1}{3} + O(k^{-1}) > 0.$$

In addition to these high-$k$ and low-$k$ asymptotic results, we can show the inequalities also hold at $k = 1$ where $\frac{1}{2}\left(\log 2 - \frac{1}{3} + 1\right) < \log 2 < \frac{1}{2}(e^{-\gamma} + 1)$, but otherwise we're relying on the graphical and numerical results. But though there is considerable margin in the asymptotes, and the median is well behaved (unique, monotonic, and smooth, with positive second derivative for all $k > 0$ [3, 5]), we do not have a proof that these are bounds. But if they are bounds, they are tight, in the sense that the positivity constraints would not all hold, since higher-order terms would not cancel, if $A_{L\infty}$ or $B_{L\infty}$ were any higher, or if $A_U$ or $B_U$ were any lower.

For the lower bound $v_{L1}(k)$ that is tight at $k = 1$, both the value and the slope need to match the true median. The value is the median of the exponential distribution, $v(1) = \log 2$. The slope $v'(k)$ is somewhat more troublesome to work out, but is tractable at the special point $k = 1$, where the CDF $P(k, x)$ (the lower incomplete gamma function) and PDF $p(k, x)$ are both exponential functions.

$$P(k, x) = \int_0^x p(k, t)dt = \int_0^x \frac{t^{k-1}}{\Gamma(k)}e^{-t}dt.$$

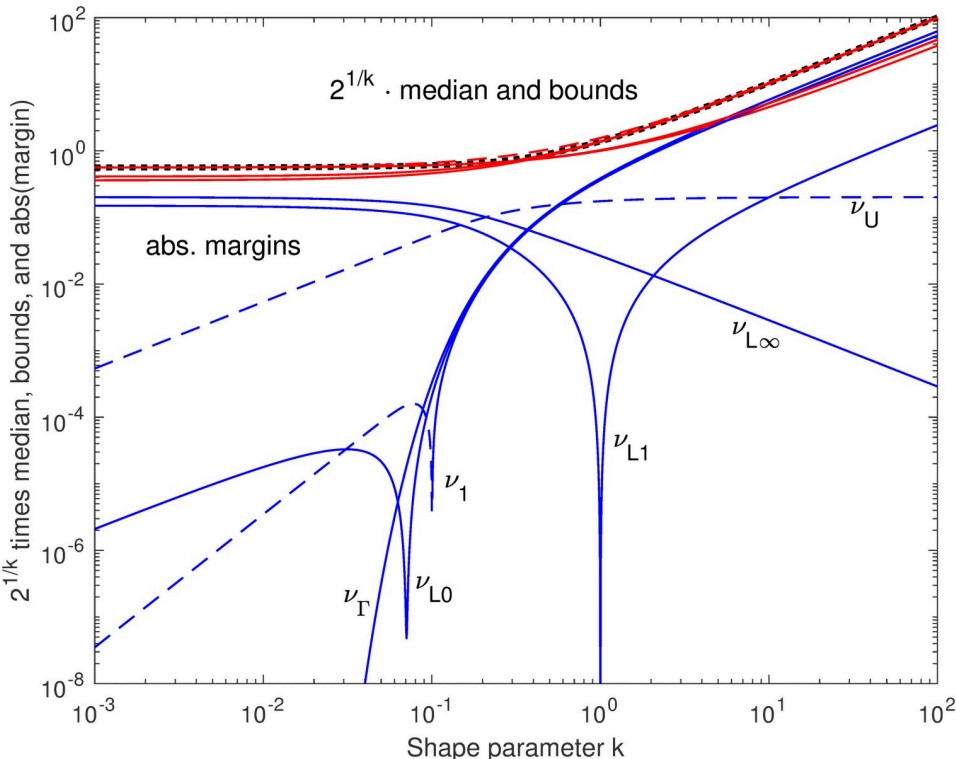

**Fig 5. New bounds and their margins, premultiplied by $2^{1/k}$.** The lower bounds $\nu_\Gamma$, $\nu_{L0}$, $\nu_{L1}$, and $\nu_{L\infty}$ (solid), and upper bound $\nu_U$ (dashed) are shown in red over the ideal median (black heavy dots), with their absolute errors in blue, all premultiplied by $2^{1/k}$ to reduce the required plot range. The approximation $\nu_1$, which is not a bound, is also shown; note that its error curve changes from solid to dashed at the cusp, while the margins for $\nu_{L0}$ and $\nu_{L1}$ have cusps (on the log scale) where the margins graze zero but do not change sign. The $k$ parameters at these cusps correspond to the sloped black lines indicated in the previous two figures.

At the point where $P(k, x) = \frac{1}{2}$, where $x = \nu(k)$, the slope is:

$$\nu'(k) = \frac{d\nu}{dk} = -\frac{\partial P(k,x)}{\partial k} \bigg/ \frac{\partial P(k,x)}{\partial x}$$

The derivative with respect to $x$ is easy, except that we only have a closed-form relation between $x$ and $k$ at $k = 1$, where we know $x = \nu(1) = \log 2$ and $p(k, x) = e^{-\log 2} = \frac{1}{2}$, so the derivative is $\frac{1}{2}$ there. The derivative with respect to $k$ is messier:

$$\frac{\partial P(k,x)}{\partial k} = -\Gamma(k)^{-2} \frac{d\Gamma(k)}{dk} \int_0^x t^{k-1} e^{-t} dt + \Gamma(k)^{-1} \int_0^x \frac{dt^{k-1}}{dk} e^{-t} dt$$

At $k = 1$, using $\Gamma(k) = 1$ and $d\Gamma(k)/dk = -\gamma$, this derivative evaluates to

$$\frac{\partial P(k,x)}{\partial k}\bigg|_{k=1} = \frac{\gamma}{2} + \int_0^x \log t \, e^{-t} dt$$

$$= \frac{\gamma}{2} + \text{Ei}(-\log 2) - \gamma - \frac{1}{2} \log \log 2$$

where $\text{Ei}(-\log 2) = -0.3786710$ is the exponential integral (integration and evaluation assisted

by Wolfram Alpha). Putting these results together we get the slope of the median:

$$v'(k)|_{k=1} = \gamma - 2\mathrm{Ei}(-\log 2) - \log \log 2 \approx 0.9680448$$

which is a mathematical expression with a definite value, but is not a closed form due to the exponential integral, so still requires a numerical approach to evaluate it. Therefore, we describe this bound with approximate numerical parameters instead of closed-form analytic expressions.

$$B = 2(v'(k)|_{k=1} - \log^2 2) \approx 0.9751836,$$

$$A = 2\log 2 - B \approx 0.4111107.$$

Unlike the straight-line lower bound $v(k) < \log 2 + (k-1)v'(1)$, which easily follows by convexity of $v(k)$, this new function $v_{L1}(k)$ that is tangent at the same point is not proved to be a bound.

The conjectured tight lower bound $v_{L0}(k)$ is in worse shape, as we have to search for the $k$ value that gives the lowest $B$ value with $A = e^{-\gamma}$. So its $B$ parameter has no concise mathematical expression, but can be computed to high precision: $B \approx 0.4596507$.

The conjectured bounds and asymptotic approximations discussed here are summarized in Table 1. In subsequent sections we focus primarily on the new upper and lower bounds with closed-form coefficients, $v_U(k)$ and $v_{L\infty}(k)$, as a basis for even tighter closed-form bounds. Fig 6 shows the percentile values achieved by these bounds and approximations, compared to the ideal 50% that defines the median: $v_{L\infty}(k)$ always comes in between 48% and 50% and $v_U(k)$ always between 50% and 55% (percentiles are calculated in Matlab as 100*gammainc(x, k), using the normalized lower incomplete gamma function that is the CDF for our PDF).

The coefficient of $k^{-1}$ for $v_{L\infty}(k)$ is negative, so the lower bound $v_{L\infty}(k)$ at high $k$ is less than the $k - \frac{1}{3}$ lower bound, in spite of having asymptotically zero absolute margin. That is, for $k > 3.021$, it's a looser lower bound and a worse approximation than $k - 1/3$, even though it is the tightest lower bound of the form we're considering. On the other hand, $v_U(k)$ is a much tighter upper bound than $k$ is, for all $k$, with an asymptotic margin near $\frac{1}{3} - (\log 2 - e^{-\gamma}) \approx 0.202$; for $k \geq 1$, the recent straight-line bound $k - 1 + \log 2$ [4] is tighter still, with an asymptotic margin $\frac{1}{3} - (1 - \log 2) \approx 0.026$.

## Formulae for tighter bounds

Letting $A$ and $B$ be functions of $k$, rather than constants, allows tighter bounding expressions (and potentially exact expressions) for $v(k)$, but not enough structure. Allowing only one of them to vary, and tying the other to values used in the tight bounds above, allows a more constrained space of bounds.

With $B_{L\infty} = B_U = 1$, we can express the median exactly as $v(k) = 2^{-1/k}(A(k) + k)$, for some smooth positive real function $A(k)$ that runs from a limit of $A_U = e^{-\gamma}$ as $k \to 0$ to $A_{L\infty} = \log 2 - \frac{1}{3}$ as $k \to +\infty$; it is apparently monotonic. Alternatively, using $A_{L0} = A_U = e^{-\gamma}$, the formula $v(k) = 2^{-1/k}(e^{-\gamma} + B(k)k)$ has a smooth positive but non-monotonic function $B(k)$ that runs between limits $\frac{e^{-\gamma}\pi^2}{12}$ and 1, but drops a little below its low-$k$ limit before increasing.

This approach converts the problem of finding tighter bounds to the median to the problem of finding closed-form expressions to bound these more well-behaved functions. Calculating $A(k)$ and $B(k)$ numerically to high precision is easy when the median can be calculated; see Fig 7. For the rest of this paper, we focus on $A(k)$, since it is monotonic and more nearly symmetric on a log $k$ axis, and because it corresponds to interpolation between closed-form bounds.

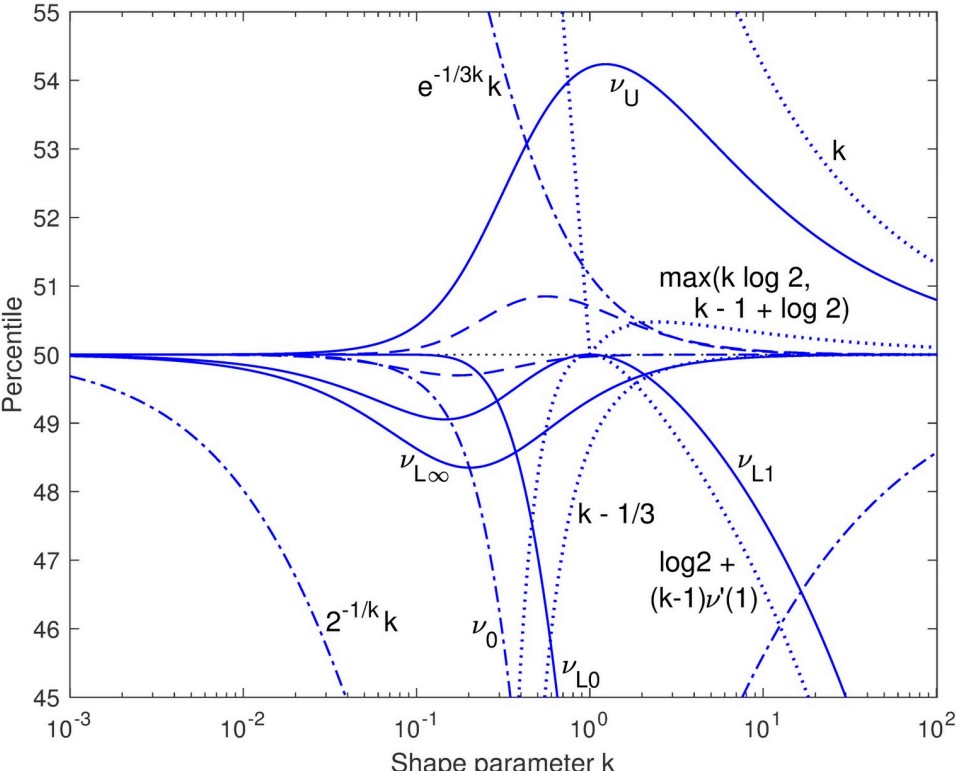

**Fig 6. Percentiles achieved.** The percentiles achieved by four conjectured median bounds of the form $2^{-1/k}(A + Bk)$ (solid curves) are plotted, along with the straight-line bounds (dotted), upper and lower bounds from Berg and Pedersen [3] (dash-dot), and a pair of closer bounds formed by interpolation between $v_U(k)$ and $v_{L\infty}(k)$ using a one-parameter rational function (dashed). The bounds $2^{-1/k} k$, $v_U(k)$, $v_{L\infty}(k)$, and the interpolated bounds converge on 50th percentile at both low and high $k$, while the other eight do not. Both the upper and lower interpolated bounds are close to $v_U(k)$ at low $k$ and close to $v_{L\infty}(k)$ at high $k$; tighter such interpolated bounds, developed in a later section, would crowd the center of the graph.

## Toward a proof

The conjectured inequalities $v_{L\infty}(k) < v(k) < v_U(k)$ are equivalent to $\log 2 - \frac{1}{3} < A(k) < e^{-\gamma}$; that is, that $A(k)$ stays above its high-$k$ limit and below its low-$k$ limit. We know that the asymptotic slopes of $A(k)$ are negative at both ends (and in the middle at $k = 1$), so it will be sufficient to show that the slope is negative everywhere; or that the function $A(k)$ is convex. It is not quite enough that $A(k)$ comes from monotonic and convex parts $2^{1/k}$ and $v(k)$. The proof of convexity of $v(k)$ [5] was complicated, and similar techniques might be needed here.

Numerically, we can see that $A(k)$ is mostly sufficiently confined between other known bounds in different parts of the $k$ range, but bounding it with other bounds could be a hard way to construct a proof. First, we'd need a better low-$k$ upper bound, which we might get using $1 - x < e^{-x}$ as we used $e^{-x} < 1$ in finding $v_\Gamma(k)$. And on the lower side we could attempt to prove that the quantile mechanics [12] partial sums are lower bounds that are tighter, at low $k$, and that the Laurent series partial sum through the $k^{-5}$ term is a tighter lower bound at high $k$.

Lacking proof of our main result, we cannot even start to prove the tighter bounds found in coming sections, based on interpolation, but again we have good confidence from asymptotic and numerical results. In some cases, asymptotic analysis yields closed-form expressions for

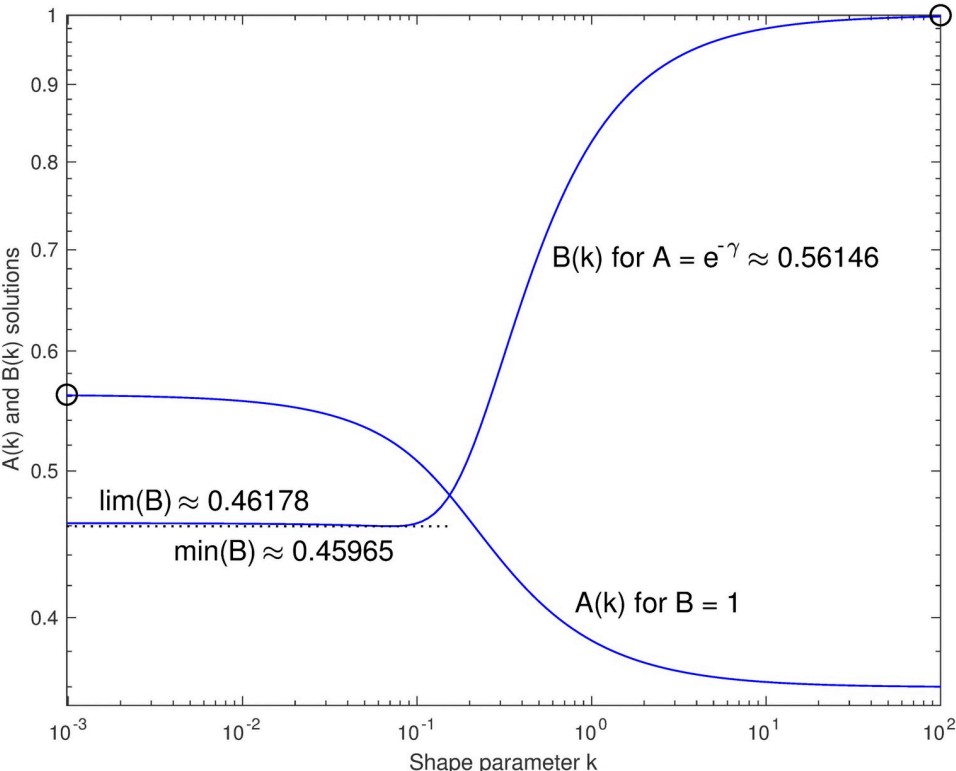

**Fig 7. Ideal parameters as functions of $k$.** Functions $A(k)$ and $B(k)$, either of which can solve $v(k) = 2^{-1/k}(A + Bk)$, with the other constant at the limiting values indicated by the circles, the parameters of $v_U$. Modeling either of these curves can lead to better bounds or approximations.

the tightest bounds of the families, while numerical methods support the conjecture that they are bounds.

## Interpolators

The function $A(k)$ introduced above can be represented in terms of an interpolation function $g(k)$ that runs monotonically from a low-$k$ limit of 0 to a high-$k$ limit of 1:

$$A(k) = g(k)A_{L\infty} + (1 - g(k))A_U = A_U - g(k)(A_U - A_{L\infty})$$

$$= e^{-\gamma} - g(k)\left(e^{-\gamma} - \log 2 + \frac{1}{3}\right).$$

And $g(k)$ is therefore also the function that interpolates between the bounds, allowing us to write the median in these convenient ways:

$$v(k) = g(k)v_{L\infty}(k) + (1 - g(k))v_U(k)$$

$$v(k) = 2^{-1/k}\left(e^{-\gamma} - g(k)\left(e^{-\gamma} - \log 2 + \frac{1}{3}\right) + k\right).$$

The ideal interpolator can be computed numerically from $A(k)$ or from $v(k)$:

$$g(k) = \frac{A_U - A(k)}{A_U - A_{L\infty}} = \frac{v_U(k) - v(k)}{v_U(k) - v_{L\infty}(k)}.$$

It can be interesting to bound or otherwise approximate $g(k)$. In approximating the ideal with an interpolator $\tilde{g}(k)$, we achieve absolute and relative error of the median estimate approaching zero at low $k$ if $\tilde{g}(k) = 0 + O(k)$, and at high $k$ if $\tilde{g}(k) = 1 - O(k^{-1})$. But we might want to do better, matching the asymptotic slopes of the ideal interpolator to match the median to a higher order; or we might want to match the exact known value $v(1) = \log 2$. So we analyze these properties of the ideal interpolator, and give them names. At low $k$:

$$P_0 = \frac{dg}{dk} = \frac{1 - \frac{e^{-\gamma}\pi^2}{12}}{e^{-\gamma} - \log 2 + \frac{1}{3}} \approx 2.66913.$$

At high $k$:

$$P_\infty = -\frac{dg}{d\frac{1}{k}} = \frac{\frac{8}{405} + e^{-\gamma}\log 2 - \frac{\log^2 2}{2}}{e^{-\gamma} - \log 2 + \frac{1}{3}} - \log 2 \approx 0.143472.$$

And at $k = 1$:

$$P_1 = g(1) = \frac{1 + e^{-\gamma} - 2\log 2}{e^{-\gamma} - \log 2 + \frac{1}{3}} \approx 0.868678.$$

How such improved-approximation goals relate to bounds is not immediately clear. In the next two sections, we construct some examples, with results summarized in Table 2. Most of the interpolated approximations and bounds listed are closed-form analytic expressions, but a few others are numeric and approximate.

**Table 2. Comparison of interpolations.**

| Version | Parameter | | |
|---|---|---|---|
| $\tilde{g}_1(k) = \frac{k}{k+b_0}$ | Symbolic $b_0$ | Numeric $b_0$ | |
| best at low $k$ | $\frac{e^{-\gamma}-\log\ 2+\frac{1}{3}}{1-\frac{e^{-\gamma}\pi^2}{12}}$ | 0.374654 | U* |
| exact at $k = 1$ | $\frac{e^{-\gamma}-\log\ 2+\frac{1}{3}}{1+e^{-\gamma}-2\ \log\ 2} - 1$ | 0.151175 | – |
| best at high $k$ | $\frac{\frac{8}{405}+e^{-\gamma}\log\ 2-\frac{\log^2 2}{2}}{e^{-\gamma}-\log\ 2+\frac{1}{3}} - \log\ 2$ | 0.143472 | L* |
| $\tilde{g}_a(k) = \frac{2}{\pi}\tan^{-1}\frac{k}{b}$ | Symbolic $b$ | Numeric $b$ | |
| best at low $k$ | $\frac{24}{\pi}\left(\frac{e^{-\gamma}-\log\ 2+\frac{1}{3}}{12-e^{-\gamma}\pi^2}\right)$ | 0.238512 | U* |
| best at high $k$ | $\frac{\pi}{2}\left(\frac{\frac{8}{405}+e^{-\gamma}\log\ 2-\frac{\log^2 2}{2}}{e^{-\gamma}-\log\ 2+\frac{1}{3}} - \log\ 2\right)$ | 0.225366 | – |
| minimax relative error | $\operatorname{argmin}_r\max\left|\frac{v(k)-\tilde{v}(k)}{v(k)}\right|$ | 0.21639 | – |
| minimax absolute error | $\operatorname{argmin}_r\max\|v(k) - \tilde{v}(k)\|$ | 0.21008 | – |
| exact at $k = 1$ | $\cot\left(\frac{\pi}{2} \cdot \frac{1+e^{-\gamma}-2\ \log\ 2}{e^{-\gamma}-\log\ 2+\frac{1}{3}}\right)$ | 0.209257 | – |
| tangent at $k \approx 0.4184$ | ? | 0.205282 | L* |

Several one-parameter interpolated bounds and approximations $\tilde{g}(k)v_{L\infty}(k) + (1 - \tilde{g}(k))v_U(k)$, some of which have closed-form parameters, are listed for comparison. Conjectured bounds are indicated by U* or L*.

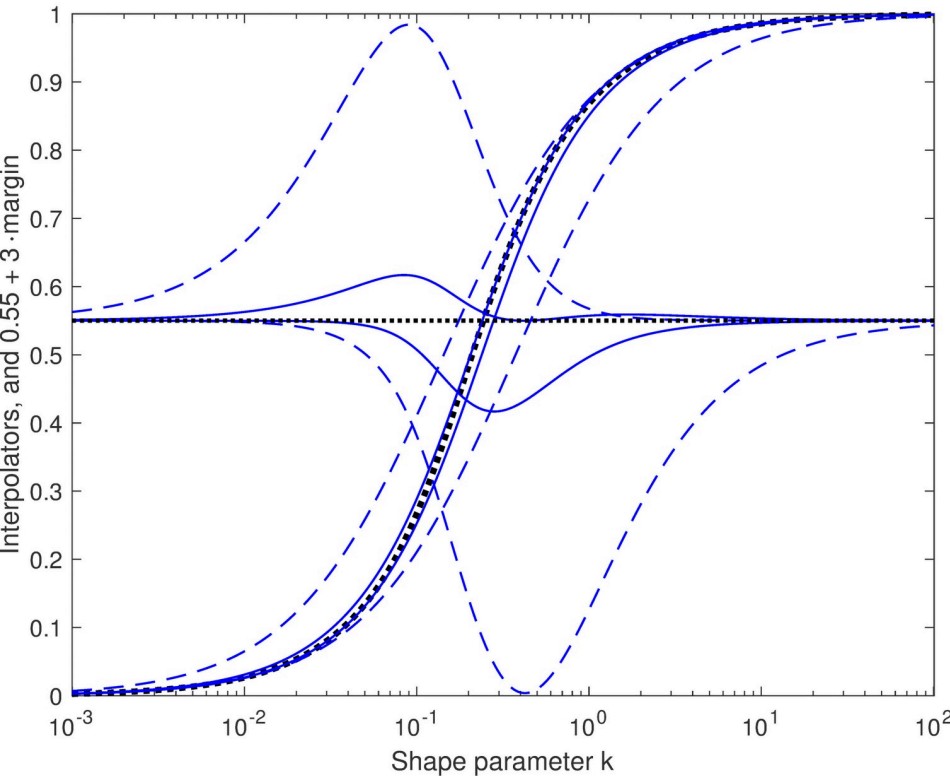

**Fig 8. Bounding the ideal interpolator function.** The ideal interpolator $g(k)$ (heavy dotted sigmoid) is compared with upper and lower bounds $\tilde{g}(k)$; their margins $\tilde{g}(k) - g(k)$ are also plotted, magnified and displaced, with the same curve styles. The curves with largest absolute margins (dashed), which correspond to the interpolated bounds shown in Fig 6, are for the first-order rational-function interpolator $\frac{k}{k+b_0}$, while the curves with smaller margins (solid) are for arctan interpolators $\frac{2}{\pi} \tan^{-1} \frac{k}{b}$. In each case, the one parameter ($b_0$ or $b$) is chosen to give a tight bound (analytically in closed form in three of the four cases). Lower bounds of $g(k)$ make upper bounds of $v(k)$, and vice versa.

Fig 8 shows the ideal interpolator, computed numerically, and compares it to bounding interpolators of the forms $\tilde{g}(k) = \frac{k}{k+b_0}$ and $\tilde{g}(k) = \frac{2}{\pi} \tan^{-1} \frac{k}{b}$, with parameters $b_0$ and $b$ chosen to yield tight upper and lower bounds. The effects of these interpolator bounds on the median bounds is shown in Fig 9.

## Rational-function interpolators

Consider rational functions as interpolators, of the form

$$\tilde{g}_N(k) = \frac{\sum_{n=1}^{N-1} a_i k^i + k^N}{\sum_{n=0}^{N-1} b_i k^i + k^N}.$$

For $N = 1$, the only parameter is $b_0$, so we have a one-parameter family:

$$\tilde{g}_1(k) = \frac{k}{b_0 + k}.$$

For $N = 2$ we have three parameters:

$$\tilde{g}_2(k) = \frac{a_1 k + k^2}{b_0 + b_1 k + k^2},$$

and so forth.

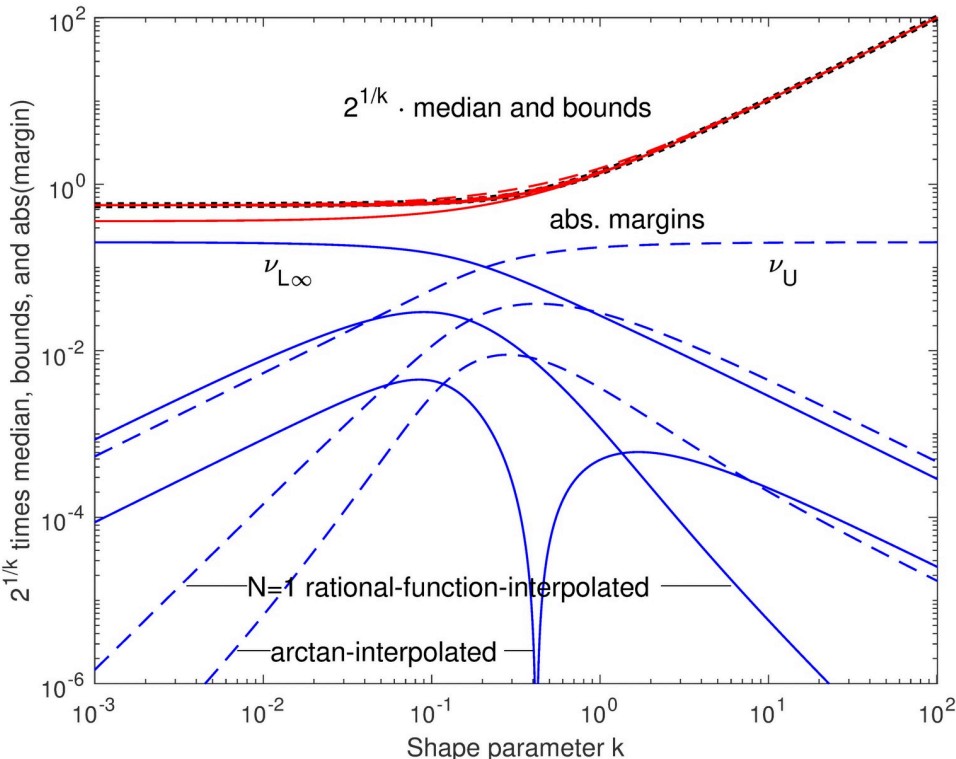

**Fig 9. Margins of interpolated bounds, premultiplied by $2^{1/k}$.** The absolute margins of the interpolated bounds are smaller than those of the bounds they started from. Compare Fig 5. The generally smallest margins are for the arctan interpolators, and the intermediate for the $N = 1$ rational-function interpolators.

We can easily constrain the coefficients to match the properties of the ideal interpolator. At low $k$:

$$\frac{a_1}{b_0} = P_0.$$

At high $k$

$$b_{N-1} - a_{N-1} = P_\infty.$$

And at $k = 1$:

$$\frac{\sum_{n=1}^{N-1} a_i + 1}{\sum_{n=0}^{N-1} b_i + 1} = P_1.$$

For $N = 1$, the low-$k$ asymptote is tightly approached with $b_0 = \frac{1}{P_0}$, yielding an upper bound for the median (lower bound for the interpolator $g_{\text{ideal}}$). Or the high-$k$ asymptote is tightly approached with $b_0 = P_\infty$, yielding a lower bound for the median (upper bound for $g(k)$). These bounds are illustrated in Fig 8. For $b_0$ between these values, the resulting interpolated function is not a bound, but is exact at one value of $k$, for example at $k = 1$ with $b_0 = \frac{1}{P_1} - 1$.

For $N = 2$, there are enough parameters to use any or all of the three constraints, but it's not immediately clear which sets of constraints can lead to bounds. Certainly using all three constraints does not lead to a bound, but to an interesting approximation. As we did with the $A$ versus $B$ space, we can investigate the locus of parameter solutions for each $k$, and examine the

edges of these locus-filled areas for tight bounds. Reducing the space to 2D by constraining one asymptote or the other allows a graphical approach, but the results are not better than a one-parameter arctan interpolator.

For $N \geq 3$, excellent approximations and bounds are possible, but with so many parameters are not very interesting. For example, we can choose to constrain both asymptotes to a higher-order fit, and to constrain the value at $k = 1$, minimizing the maximum relative error with the two remaining parameters. It's an excellent fit in all regions, with maximum relative error of 0.00018, but could be even better in the middle without the constraints:

$$\tilde{g}_3(k) = \frac{0.019983\,k + 0.083933\,k^2 + k^3}{0.0074867 + 0.0359083\,k + 0.227405\,k^2 + k^3}$$

## Arctan interpolators

Getting a tighter bound or better approximation from the rational-function family requires at least a handful of parameters. An alternative approach is to find a one-parameter shape that fits better. We have found that the arctan shape with parameter $b$ (like $b_0$ in the one-parameter rational-function interpolator, corresponding to the $k$ value at the midpoint, $\tilde{g}(b) = 0.5$) does a good job:

$$\tilde{g}_a(k) = \frac{2}{\pi}\tan^{-1}\frac{k}{b}.$$

As Fig 8 shows, the shapes of the arctan interpolators are imperfect, but are much better than the first-order rational function, fitting better in some regions than in others. Note that both the $N = 1$ rational function and the arctan interpolators are symmetric about their centers (with $\log k$ as the independent variable); they are logistic sigmoid and Gudermannian shapes, respectively. The ideal that they are to bound or approximate, however, is not quite symmetric in $\log k$. So a family of not-quite-symmetric interpolators can perhaps do better.

Several special $b$ parameter values for the arctan interpolator can be derived to match each of the ideal properties mentioned above. We can constrain the approximation to pass through the known value $v(1) = \log 2$, with $b = \cot\left(\frac{\pi}{2}P_1\right)$, but that does not give a bound. Or we match the true median at high $k$ to within $O(k^{-2})$ with $b = \frac{\pi}{2}P_\infty$. That also does not give a bound. Or we can match at low $k$ to within $O(2^{-1/k}k^2)$, like $\tilde{v}_1$ does, with $b = \frac{2}{\pi}P_0^{-1}$, which yields a lower bound to $g$.

To find an upper bound to $g$ (lower bound to $v$), we decrease $b$ until the margin is nonnegative for all $k$, which is at about $b = 0.205282$. Finding an analytic formulation for that tight bound is a challenge left to others.

See Fig 10 for the relative errors of various arctan interpolator versions.

## Conclusions

Tight upper and lower bounds to the median of the gamma distribution are conjectured, based on numerical and asymptotic analyses. The simplest conjectured lower bound is never below the 48th percentile, and the simplest conjectured upper bound, of the same form $2^{-1/k}(A + Bk)$, is never above the 55th percentile, over the entire range of $k > 0$. Using arctan and rational-function interpolators between these closed-form bounds, two better one-parameter families of bounds and approximations to the median of a gamma distribution are proposed.

The one-parameter rational-function family has simple closed-form formulae for tightest upper and lower conjectured bounds, staying below 50.85 and above 49.69 percentile, respectively; higher-order rational functions can provide tighter bounds or better approximations.

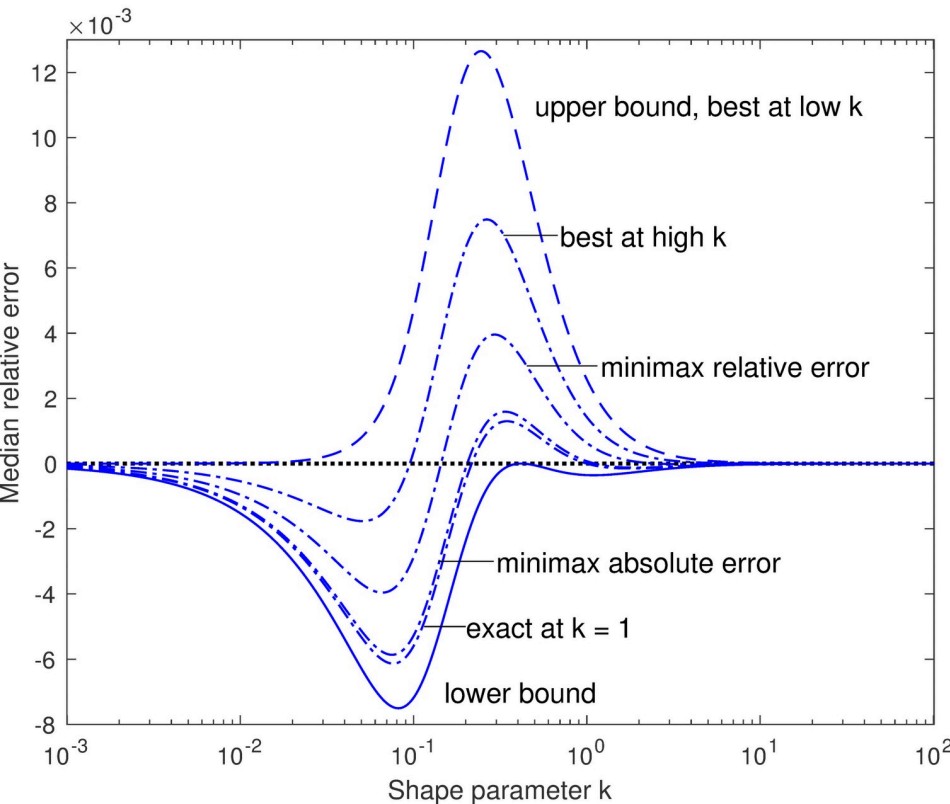

**Fig 10. Relative errors of arctan interpolations.** Relative errors of arctan-interpolated approximations (dash-dot curves) between the upper (dashed) and lower (solid) bounds. These are among the possibly interesting approximations suggested by the tight-bounds approach. The non-bounding approximations, optimized for different criteria, all have maximum relative errors below 1%.

The one-parameter arctan family of interpolators is a better fit to the ideal interpolator, and includes a version that is most accurate in the low-$k$ tail and provides a closed-form tight con-jectured upper bound, staying below 50.18 percentile. With different $b$ parameter, several approximations in the family, including the closed-form version that is exact at $k = 1$, stay between 49.97 and 50.03 percentile. We have not found an analytic formula for the parameter that gives the tightest lower bound, which stays above 49.96 percentile, but have shown where to find it, graphically or numerically.

The approach of interpolating between tight bounds opens the way to finding tighter bounds and more accurate approximations, and to finding more such families of bounds and approximations via other interpolator forms.

While numerical and graphical techniques were used in finding these bounds, the ones with closed forms are grounded in asymptotic analysis. Proving that they are in fact bounds remains an unmet challenge.

## Acknowledgments

The author gratefully acknowledges helpful comments on previous drafts from Google col-leagues Pascal Getreuer, Srinivas Vasudevan, Dan Piponi, and Michael Keselman, and from outside experts D. Andrew Barry, José Antonio Adell, Milan Merkle, Christian Berg, and Hen-rik L. Pedersen. The anonymous reviewers were also very helpful.

## Author Contributions

**Conceptualization:** Richard F. Lyon.

**Formal analysis:** Richard F. Lyon.

**Investigation:** Richard F. Lyon.

**Methodology:** Richard F. Lyon.

**Software:** Richard F. Lyon.

**Visualization:** Richard F. Lyon.

**Writing – original draft:** Richard F. Lyon.

**Writing – review & editing:** Richard F. Lyon.

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
