## [Decision Letter · Decision Letter 0]

15 Mar 2021

PONE-D-21-00917

Closed-form tight bounds and approximations for the median of a gamma distribution

PLOS ONE

Dear Dr. Lyon,

Thank you for submitting your manuscript to PLOS ONE. After careful consideration, we feel that it has merit but does not fully meet PLOS ONE’s publication criteria as it currently stands. Therefore, we invite you to submit a revised version of the manuscript that addresses the points raised during the review process.

We look forward to receiving your revised manuscript.

Kind regards,

Ivan Kryven

Academic Editor

PLOS ONE

Journal Requirements:

We note that one or more of the authors are employed by a commercial company: Google Inc.

2.1. Please provide an amended Funding Statement declaring this commercial affiliation, as well as a statement regarding the Role of Funders in your study. If the funding organization did not play a role in the study design, data collection and analysis, decision to publish, or preparation of the manuscript and only provided financial support in the form of authors' salaries and/or research materials, please review your statements relating to the author contributions, and ensure you have specifically and accurately indicated the role(s) that these authors had in your study. You can update author roles in the Author Contributions section of the online submission form.

2.2. Please also provide an updated Competing Interests Statement declaring this commercial affiliation along with any other relevant declarations relating to employment, consultancy, patents, products in development, or marketed products, etc.  

Reviewers' comments:

Reviewer's Responses to Questions

**Comments to the Author**

1. Is the manuscript technically sound, and do the data support the conclusions?

Reviewer #1: Yes

Reviewer #2: Yes

2. Has the statistical analysis been performed appropriately and rigorously? 

Reviewer #1: N/A

Reviewer #2: N/A

3. Have the authors made all data underlying the findings in their manuscript fully available?

Reviewer #1: Yes

Reviewer #2: Yes

4. Is the manuscript presented in an intelligible fashion and written in standard English?

Reviewer #1: Yes

Reviewer #2: Yes

5. Review Comments to the Author

Reviewer #1: Please see the attached report *******************************************************************************************************************************************************************************

Reviewer #2: The topic of the manuscript is very important in mathematics and , due to the role of median in various problems. The median of Gamma distribution with a fixed shape parameter k can't be represented in terms of elementary functions, so

the the idea of the author to use approximation based on two parameters A and B, of type as in the page 3, is very convenient.

Some comments:

1) First sentence in Abstract: Delete "The tightest possible", as this is not proved by mathematics. Further,

in place of "interesting cases" describe numerically what are those cases.

2) Text under Fig 3. Parameter space detail. It seems that the " (see the next Fig)" is a mistake?

6. PLOS authors have the option to publish the peer review history of their article (what does this mean?). If published, this will include your full peer review and any attached files.

Reviewer #1: No

Reviewer #2: No

---

## [Author Response · Author response to Decision Letter 0]

2 Apr 2021

Rebuttal Letter / Response to Reviwers

by Richard F. Lyon

[PONE-D-21-00917]

31 March 2021

I appreciate the positive reviews, including reviewer 1's "tough love" approach to explaining how I need to improve the presentation for conjectured bounds.  I mostly made changes in line with their suggestions, and added a few more bits along the way to help clarify (such as the new section "Toward a proof" that might help others complete the work, and an example of a third-order rational function interpolation with low relative error).

First, just two changes per reviewer 2's two suggestions:

in abstract:

The tightest possible upper and lower bounds of the form $2^{-1/k} (A + Bk)$ to the median of a gamma distribution, over the entire range of shape parameter $k > 0$, have closed-form parameters $A$ and $B$ in interesting cases. 

-- changed to 

Conjectured tight upper and lower bounds of the form $2^{-1/k} (A + Bk)$ to the median of a gamma distribution over the entire range of shape parameter $k > 0$ have closed-form parameters $A$ and $B$ in the case of the lower bound that is tight for high $k$ and the upper bound that is tight for low $k$.  

in Fig 4 caption: (previously Fig 3)

(see the next Fig) 

--was not wrong, but unclear; changed to 

(see their errors, or margins, in Fig. 5)

Then back to reviewer 1's lengthier comments:

General Comments

(1) Throughout this report, I have been careful to stress that most of the author’s bounds are conjectured. This is something the author needs to do more clearly. I acknowledge that the author does put an asterisk to denote bounds that have not been rigorously proved and writes in the conclusion “Proving that they [the conjectured bounds] are in fact bounds remains an unmet challenge.” However, the point that the bounds are conjectured needs to be made very clear early in the paper. Indeed, it wasn’t until I reached page 4 of this short paper that I realised that the bounds were not going to be rigorously proven, which is what one expects when reading a paper on mathematics. It is important that the reader is made aware at an early stage that the bounds are conjectured. For example, if another researcher uses the author’s bounds to bound a quantity themselves then the bound they obtain is in turn non-rigorous, and they must be made aware of this. The author therefore needs to make the following edits. The title must change. An example would be “Approximations and conjectured closed-form tight bounds for the median of a gamma distribution”, or if the author prefers not to use the word ‘conjecture’ in the title, something like “On closed-form tight bounds and approximations for the median of a gamma distribution”. The author must clearly state in the Abstract that the bounds are conjectured, and can mention that they are supported by asymptotic, numerical and graphical arguments. It must be mentioned in the Introduction that the bounds are conjectured. The first sentence of the Conclusion should be edited from “Tight upper and lower bounds to the median of the gamma distribution are introduced” to something like “Conjectured tight upper and lower bounds to the median of the gamma distribution are introduced”. Moreover, the author may wish to modify certain sentences throughout the paper to reflect the more modest findings of this research. For example, the sentence on page 5 that ends with “this seems reliable enough in concluding that these are bounds (but mathematicians are invited to interpret these as conjectures to be proved)” needs to be edited. There is no doubt that the bounds are conjectures to be proved.

Added "On" in the title, and "conjectured" in abstract, introduction, conclusion, and lots of places in between; took out the bit about "seems reliable enough" and instead wrote a short section "Toward a proof" to discuss a couple of approaches that might work.

(2) On page 2, the author presents the well-known two-sided inequality of Chen and Rubin for the true mean ν of a gamma distribution with rate parameter ν and scale parameter 1: k−1/3 < ν < k. Throughout the paper the author 3 compares his bounds to these bounds. However, for k ≥ 1, the upper bound can be significantly improved: Gaunt and Merkle [1, Theorem 3.1] give the upper bound ν < k − 1 + log(2), k > 1; note that there is equality at k = 1, which corresponds to the case of the exponential distribution. This upper bound extends the range of validity of a previous bound of Choi (reference 3) from positive integer k to all k > 1. As 1 − log(2) ≈ 0.306852 is ‘close’ to 1/3, when this upper bound is combined with the lower bound k − 1/3 it results in a very accurate two-sided inequality for the median. The author should report this bound of [1] in either the ‘Problem formulation’ or ‘Prior work’ sections. The author should also update their numerical results that give comparisons to existing bounds to include a comparison to this bound. As this upper bound is very accurate, some of the author’s conclusions regarding the performance of his bounds relative to others in the literature may change. The paper [1] (see Section 3) also gives conjectured inequalities for the median of the variance-gamma distribution. In particular, the authors conjecture that the median of the variance-gamma distribution can be bounded above and below by the median of certain gamma distributions. Therefore the author’s conjectured accurate bounds may in turn, at least for some parameter values, result in improved conjectured bounds for the median of variance-gamma distribution. I will leave up to the author to decide whether they wish to add such a remark to their paper.

Thanks for that better bound.  I have added a brief discussion of the 2021 Gaunt & Merkle bound in the prior art section.  Since it is not a bound over the entire domain k > 0, I extended it with a chord to make a complete piecewise linear bound, and made a new Fig. 1 to show it along with linear bounds.  I also added that piecewise-linear boound to several other figures, and discussed the fact that it has much lower margin at high k than my new upper bound.

The variance-gamma distribution is outside what I understand, so I'll leave it out of scope for this paper.

Specific Comments

(1) Page 1, Abstract (and elsewhere): The author uses “closed-form” and closed form. Please consistently use just one of these. 

The current usage is consistent with the advice of many English style and grammar guides that suggest hyphenating a compound noun when used as a modifier, but not otherwise, as in:

  Closed-form tight bounds

  closed-form expressions

  closed-form parameters

  closed-form bounds

  closed-form analytic expressions

  closed-form tight upper bound (was misspelled closed-from tight upper bound)

  closed-form version

  in closed form (as a noun phrase, not a modifier, no hyphen should be used)

example of a paper that adheres to this distinction:  https://psi.ece.jhu.edu/kaplan1/PUBL/AEK.pubs/104.pdf

advice in guides:

  https://en.wikipedia.org/wiki/Compound_modifier#Hyphenation_of_elements_in_English

  https://www.grammarly.com/blog/hyphen/

  https://www.grammarly.com/blog/hyphen-with-compound-modifiers/

  https://www.constant-content.com/content-writing-service/2014/01/compound-modifiers-the-easy-to-use-hyphen/

  https://sites.utexas.edu/legalwriting/2014/07/21/the-compound-modifier-hyphen-connects-and-clarifies/

  https://blog.apastyle.org/apastyle/2016/10/hyphenation-station-using-compound-adjectives.html

Extended discussion illustrating that not everyone agrees, but I and several other prominent Wikipedia style gurus do:

  https://en.wikipedia.org/wiki/Wikipedia_talk:Manual_of_Style/Archive_211#Specialized-style_fallacy_and_resistance_to_hyphenation_of_compound_adjectives

If PLOS One has a preferred hyphenation style, I will follow it; but for now, I think the consistent style that I have followed should be OK.  It agrees, for example, with the Cambridge University Press style that I used in my book.

(2) Page 1, Introduction: I suggest the author brings Tables 1 and 2 to the reader’s attention in the introduction. These tables are very useful and efficiently display the main findings of this work. A reader who wants to quickly get to the results would find such a comment in the introduction to be very helpful. 

I'm glad the tables were as helpful as I had hoped they would be.  I moved Table 1 reference to much earlier (but not in the introduction).  And Table 2 somewhat earlier, too, so they summarize what's coming instead of what's done.  I mention "the tables" in the introduction, but not by number as that would require me to move them up there.

(3) Page 1, At the bottom of the page, the author introduces the notation ν for the median of the gamma distribution with rate parameter 1 and scale parameter k. I suggest modifying this notation to ν(k) to emphasis the dependence on k. Later in the paper, the author sometimes does write ν(k), and it would be helpful to keep things consistent throughout the paper. The same comment applies to the author’s notation for other bounds and approximations: e.g., νL∞ becomes νL∞(k). 

Yes, good idea.  I've added the "(k)" throughout to make clear that the median and the bounds are functions of k.  Hopefully I didn't miss any.

(4) Page 3: The author neatly derives the bound ν > 2 −1/kΓ(k + 1)1/k. I appreciate that the author derives this bound with the intention of motivating further work in the paper. Nevertheless, it would be helpful if the author were to give a statement as to whether they believe this to be a new result. 

Yes, the table says it's a new lower bound.  But maybe it's anticipated or implied by Quantile Mechanics.  I clarified that I'm claiming it's new even though it may be implicit in Steinbrecher and Shaw.

(5) Page 4, line −3: log(x) is not defined at x = 0, so it makes no sense to write log(0). The author therefore may consider formulating “log(0) cusps” differently.

Good point.  I meant a cusp that comes from the plot trying to reach the log of nearly 0.  Clarifying...

(6) Page 5, line −9, “The slope v'(k)”: Does the author mean \\nu'(k)? Please look out for other such typos, e.g., in the displayed equation below. Also, it would be helpful to the reader if the author were to provide an explanation or short calculation to verify how they found the formula for v'(k)|k=1. 

Fixed typo.  Changed $v^{\\prime }(k)$ to $\\nu^{\\prime }(k)$ in 3 places near there.

Derivation of the slope v'(k)|k=1 has been added.

(7) Page 8: In the final displayed equation, the comma should be a full stop.

I'm not sure I see the right place.  In the compiled submission PDF, page 8/11 concludes with "and so forth." after the comma.

---

## [Decision Letter · Decision Letter 1]

26 Apr 2021

PONE-D-21-00917R1

On closed-form tight bounds and approximations for the median of a gamma distribution

PLOS ONE

Dear Dr. Lyon,

Thank you for resubmitting your manuscript to PLOS ONE. The referee have evaluated your manuscript again and they now are positive about publishing it at our journal. Note that one of the reviewers, has two minor suggestions. Before we can proceed with formal acceptance, we invite you to consider the following editorial comments regarding the style:

**Editorial comments**

1. PLOS one has a broad audience that includes mathematicians as well as scientists from diverse areas. Please consider rewriting abstract to be more comprehensible to non-experts. For example, you may consider the following:

The median of Gamma distribution with a fixed shape parameter k can't be represented in terms of elementary functions. In this work we use numerical simulations and asymptotic analyses to bound the median, suggesting an upper bound that is tight for low k and a lower bound that is tight for large k. These bounds have the form 2−1/k(A + Bk) and are valid over the entire range of k > 0. Furthermore, an interpolation between these bounds yields closed-form expressions that tightly bound the median, with absolute and relative margins approaching zero at both low and high values of k. Some of our results are not supported with analytical proofs but are only confirmed with numerical calculations, and hence should be regarded as conjectures in the strict mathematical sense.

2. The current Introduction section is too short and not much more informative than the abstract. Consider the following: a) Removing the current Introduction. b) Renaming Problem Formulation -> Introduction,and c) joining this section with Prior Work. In which case the paragraph starting with "We seek upper and lower bounds that are tighter..." should be placed at the end of the section. 3. Following up on out previous discussion, you may include the proof of the statement "Γ(k + 1)1/k increases monotonically from e−γ, ..."

We look forward to receiving your revised manuscript.

Kind regards,

Ivan Kryven

Academic Editor

PLOS ONE

Journal Requirements:

Reviewers' comments:

Reviewer's Responses to Questions

**Comments to the Author**

1. If the authors have adequately addressed your comments raised in a previous round of review and you feel that this manuscript is now acceptable for publication, you may indicate that here to bypass the “Comments to the Author” section, enter your conflict of interest statement in the “Confidential to Editor” section, and submit your "Accept" recommendation.

Reviewer #1: All comments have been addressed

Reviewer #2: All comments have been addressed

2. Is the manuscript technically sound, and do the data support the conclusions?

Reviewer #1: Yes

Reviewer #2: Yes

3. Has the statistical analysis been performed appropriately and rigorously? 

Reviewer #1: Yes

Reviewer #2: N/A

4. Have the authors made all data underlying the findings in their manuscript fully available?

Reviewer #1: Yes

Reviewer #2: Yes

5. Is the manuscript presented in an intelligible fashion and written in standard English?

Reviewer #1: Yes

Reviewer #2: Yes

6. Review Comments to the Author

Reviewer #1: Please see the attached report *****************************************************************************************

Reviewer #2: The topic of the manuscript is very important in mathematics and , due to the role of median in various problems. The median of Gamma distribution with a fixed shape parameter k can't be represented in terms of elementary functions, so

the the idea of the author to use approximation based on two parameters A and B, of type as in the page 3, is very convenient.

7. PLOS authors have the option to publish the peer review history of their article (what does this mean?). If published, this will include your full peer review and any attached files.

Reviewer #1: No

Reviewer #2: No

---

## [Author Response · Author response to Decision Letter 1]

29 Apr 2021

Rebuttal Letter / Response to Reviewers

PONE-D-21-00917R1

On closed-form tight bounds and approximations for the median of a gamma distribution

PLOS ONE

Richard F. Lyon, author

I thank the reviewers again, and the editor, for the constructive suggestions and positive reaction to my article.  I have incorporated the latest suggestions, pretty nearly, and a few minor corrections.

Editor suggests a rewrite of the Abstract.  I did that, starting with his suggested opening but then with some changes.  Instead of "with a fixed shape parameter k" I said "as a function of shape parameter k".  I moved the algebraic form of the bounds up one sentence to near the bounds it most particularly applied to.  And I added a few more words to clarify things.

Editor suggests dropping the short Introduction paragraph and merging the next two sections into an Introduction.  Done; exactly as suggested, except also moved the sentence saying that results are summarized in tables to come, since the reviewers had asked for that.

Editor says an added proof that "Γ(k + 1)1/k increases monotonically from e−γ, ..." would be welcome, so I added that.  

Reviewer 1 pointed out notation error in >= relation.  I found and fixed 2 of those.

Reviewer 1 pointed out that I failed to add the "(k)" in a few places.  I found and fixed 9 of those (including 5 in table 2).

Reviewer 2 had no specific comments.

Other things I changed:

Fig 2 had an error in the y axis label, so I fixed that.

I  modified the comment on the potential use of quantile mechanics partial sums as bounds, as they are not closed form as I had suggested.

I re-ordered references to match the change order of citation in the new introduction.

I added an acknowledgement name, for the colleague who helped me with the proof.

That's all.

---

## [Editor Report · Decision Letter 2]

30 Apr 2021

On closed-form tight bounds and approximations for the median of a gamma distribution

PONE-D-21-00917R2

Dear Dr. Lyon,

We’re pleased to inform you that your manuscript has been judged scientifically suitable for publication and will be formally accepted for publication once it meets all outstanding technical requirements.

*Editorial comment:* When preparing the final version of the paper for production please note that the paper has multiple one-sentence paragraphs. For example, as in line 5, 7, 9 29.  Consider merging these paragraphs with the preceding text.

Kind regards,

Ivan Kryven

Academic Editor

PLOS ONE
---

## [Editor Report · Acceptance letter]

4 May 2021

PONE-D-21-00917R2 

On closed-form tight bounds and approximations for the median of a gamma distribution 

Dear Dr. Lyon:

I'm pleased to inform you that your manuscript has been deemed suitable for publication in PLOS ONE. Congratulations! Your manuscript is now with our production department. 

Kind regards, 

on behalf of

Dr. Ivan Kryven 

Academic Editor

PLOS ONE